# Clustering Analysis, Structure Fingerprint Analysis, and Quantum Chemical Calculations of Compounds from Essential Oils of Sunflower *(Helianthus annuus* L.) Receptacles

**DOI:** 10.3390/ijms231710169

**Published:** 2022-09-05

**Authors:** Yi He, Kaifeng Liu, Lu Han, Weiwei Han

**Affiliations:** Key Laboratory for Molecular Enzymology and Engineering of Ministry of Education, School of Life Science, Jilin University, 2699 Qianjin Street, Changchun 130012, China

**Keywords:** chemical space network, sunflower (*Helianthus annuus* L.), essential oils, fingerprint, clustering

## Abstract

Sunflower (*Helianthus annuus* L.) is an appropriate crop for current new patterns of green agriculture, so it is important to change sunflower receptacles from waste to useful resource. However, there is limited knowledge on the functions of compounds from the essential oils of sunflower receptacles. In this study, a new method was created for chemical space network analysis and classification of small samples, and applied to 104 compounds. Here, t-SNE (t-Distributed Stochastic Neighbor Embedding) dimensions were used to reduce coordinates as node locations and edge connections of chemical space networks, respectively, and molecules were grouped according to whether the edges were connected and the proximity of the node coordinates. Through detailed analysis of the structural characteristics and fingerprints of each classified group, our classification method attained good accuracy. Targets were then identified using reverse docking methods, and the active centers of the same types of compounds were determined by quantum chemical calculation. The results indicated that these compounds can be divided into nine groups, according to their mean within-group similarity (MWGS) values. The three families with the most members, i.e., the d-limonene group (18), α-pinene group (10), and γ-maaliene group (nine members) determined the protein targets, using PharmMapper. Structure fingerprint analysis was employed to predict the binding mode of the ligands of four families of the protein targets. Thence, quantum chemical calculations were applied to the active group of the representative compounds of the four families. This study provides further scientific information to support the use of sunflower receptacles.

## 1. Introduction

Sunflower (*Helianthus annuus* L.) belongs to the Compositae family (Asteraceae), which originated in South America and spread to China in the seventeenth century [1,2,3]. Sunflower has been widely cultivated in northeast China. After the seeds are used for oil extraction, sunflower receptacles have largely been discarded [4,5,6], which not only wastes resources, but also pollutes the environment. Sunflower receptacles contains many active compounds, including flavonoids [7,8], alkaloids [9,10], and chlorogenic acid [11,12]. However, there are few existing studies on the essential oil of sunflower receptacles. Therefore, it is in line with new pattern of green agriculture to change discarded sunflower receptacles from waste into a valuable resource.

Essential oils of sunflower are rich in unsaturated fatty acids, such as oleic and linoleic acids (ω-6), and are considered good for human health [13]. Essential oils of sunflower receptacles can be obtained by hydrodistillation [14].

In our previous studies, 101 compounds from the essential oils of sunflower (*Helianthus annuus* L.) receptacles were identified by gas chromatography-mass spectrometry (GC-MS) from three varieties of sunflowers, i.e., LD5009, SH363, and S606 [15,16]. The results showed that eupatoriochromene may be one of the most important chemical compounds of sunflower receptacles for reducing uric acid. However, the functions of the other compounds of essential oils from sunflower receptacles remain unknown.

Chemical space network (CSN) techniques map chemical molecules into a visual space according to certain characteristics including molecular structure. CSN was initially designed as a coordinate-free threshold network using the Tanimoto coefficient as a continuous similarity measure [17]. However, it can only reflect the connection between molecules and not the relative distances of molecules in space. Several studies have proposed more novel CSNs, such as the TV-CSN [18] and Kamada–Kawai network [19], the latter attempting to introduce coordinates into the CSN. Compared to the traditional threshold CSN, in this study we added t-SNE dimensionality reduction to determine the spatial coordinates, to adapt to component clustering in mixtures with large differences in composition, such as essential oils.

Protein–ligand interaction fingerprints (IFPs) are binary one-dimensional representations of the three-dimensional structures of protein–ligand complexes, encoding the presence or absence of specific interactions between the binding pocket amino acids and the ligand. IFPs have successfully been applied for post-processing molecular docking results for G protein-coupled receptor (GPCR) ligand binding mode prediction and virtual ligand screening [20].

The purpose of this study was to compare the differences in the chemical compounds of essential oils from sunflower receptacles. Over 100 compounds were clustered by mapping into a CSN, and representative compounds from each group were selected. The target molecules of each group were identified using reverse docking to the group’s representative compound. The active centers of the same type of compounds were determined by quantum chemical calculation. This study can provide reliable clues for the application of these compounds, and further scientific information to support the use of sunflower receptacles, which can reduce the waste of sunflower receptacles and increase the incomes of farmers.

## 2. Results

### 2.1. Cluster Analysis

The molecular similarities are shown for 104 compounds (Figure 1). Due to the complex composition of essential oils, similarity between molecules was found to be generally low, but the similarities between some molecules were obvious, as in the long-chain compounds (the upper left corner of the heatmap). A whitish area was observed in the heatmap (hierarchical clustering is shown in red), representing a large cluster of endocyclic compounds. Due to the special structure of ceertain molecules, their similarities with the other 103 molecules are all less than 0.45, so they did not participate in network generation. Finally, 91 of the 104 compounds had at least one edge with a similarity greater than 0.45, indicating their participation in the network generation.

The two-dimensional coordinates of 104 molecules calculated by t-SNE dimensionality reduction can be viewed in Appendix A, and were used to help generate the CSN.

According to the edge data in Figure 1 and the node coordinate data in Appendix A, we grouped molecules based on similarity greater than 0.45 and proximate node location. After filtering out groups whose total number of molecules was less than three, 73 of 91 compounds remained clustered into nine classes; see Figure 2 and Appendix A. We can see that many related molecules were not grouped together because they were too far apart, which is exactly as expected after refining the groups. The α-pinene group, d-limonene group, α-muurolene group, and γ-maalineen group were more or less connected; traditional methods have difficulty distinguishing molecules that link multiple groups, but after introducing the coordinates obtained by dimensionality reduction, the four groups were visually separated. This offered preliminarily proof that our method is feasible for analysis of essential oils with large differences in components.

Next, we analyzed the structural characteristics of each group classification. The linoleic acid group mainly comprised long chain compounds. The α-Pinene group compounds were found to be mainly bicyclic monoterpenes and their oxygen-containing derivatives. Many structural types of bicyclic monoterpenes exist, including pinene, campene, carene, and others. Among these, the pinene and camphene types are the most stable. The first nine examples in Appendix A are pinene type, with a bridged ring skeleton of 2,6,6-trimethylbicyclo [3.1.1] heptane, pinene, rosinol, myrtenol, and verbenol. All are typical pinene-type compounds. The last example is a camphene type, with a bridged ring skeleton of 1,7,7-trimethylbicyclo [2.2.1] heptane. The camphene type compounds mostly exist as oxygen-containing derivatives, such as 6-camphenone. D-limonene group compounds are mainly monocyclic monoterpenes and their derivatives including formates, ketones, alcohols, except cis-Australinol, β-bisabolene, two monocyclic sesquiterpenes, and enols. γ-Maalineen group compounds are mainly tricyclic sesquiterpenes and their oxygen-containing derivatives, including acetates, lactones, and alcohols, except for trans-valerian terpene alcohol acetates, which are acetates of bicyclic sesquiterpenes. The compounds of epimanoly oxide group are mainly oxygen-containing derivatives of tricyclic or tetracyclic diterpenes, and include acids, alcohols, ketones, formate esters, and ethers. The group can be divided into four categories. The first category is tetracyclic diterpene with kaurine diterpenes as the core skeleton, like kauri aldehyde, kauri acid, H-Kauran-16-ol, and enantio–kaurane diterpenes. They are natural products with many important biological activities including antibacterial, anti-inflammatory, and anti-tumor effects [21]. The second category is ethyl isopimaric acid with tricyclic diterpene–pimarane diterpene as the core skeleton. The third category is ribenone and 13-epimanoyl oxide with tricyclic diterpenes–helichryllane diterpenes (sclareolide) as the core skeleton. Finally, the ricyclic-diterpenes and long-leaf aldehydes have more complex bridged ring structures. The compounds of the benzene–butoxymethyl group are alcohols, ethers, and dibutyl esters containing benzene rings. The compounds of trans-sabinol group compounds are bicyclic monoterpene compounds and their oxygen-containing derivatives, including alcohols and formate esters, all of which have bicyclic structure of 4-methyl-1-isopropylbicyclo [3.1.0] hexane. The compounds of desmethoxtencecalin share the structure of benzo-α-pyran, i.e., α-chromene.

In this study, the three families with the most members (the d-limonene (18), α-pinene (10), and γ-maaliene groups (nine) were designated the protein targets for further study. Although the linoleic acid group had 10 compounds, they have been found in many plants and well researched [22], so the linoleic acid group was not considered in the next study.

### 2.2. Reverse Docking, Structure Fingerprint, and Quantum Chemical Calculation Analysis

#### 2.2.1. d-Limonene Group

There were 18 compounds in the d-limonene group, including d-limonene, the representative compound of the group (MWGS value 0.42), as shown in Appendix A. The greater the MWGS value, the more closely a compound is related to other compounds in this family, and this was used to designate the predicted target protein in PharmMapper [23,24]. The predicted target was human placental estrone/DHEA sulfatase (ES, PDB ID is 1P49), which catalyzes the conversion of sulfated steroid precursors such as dehydroepiandrosterone sulfate (DHEA-S) and estrone sulfate to the free steroid [25]. Estrone sulfatase (ES) is one of the key enzymes involving in maintaining high levels of estrogen in breast tumor cells. The presence of ES in breast carcinomas has been related to breast cancer and X-linked ichthyosis, a disease of the skin [26]. Figure 3A shows the LUMO (lowest unoccupied molecular orbital) orbits of the d-limonen. LUMO is important for the establishment of the chemical bond and is integral in the sphere of spectroscopy. It depends on all coordinates of a system, providing a more efficient sampling method than a geometrical reaction coordinate, to better reflect the activities of the compound. It can be seen that the LUMO is concentrated on the propylene group, which will obtain electrons more easily and become chemically more active.

Figure 3B,C show d-limonene group molecules binding in the active pocket of the ES, and d-limonene interacting with amino acid residues of ES. Subsequently. The interactions of the receptor to ligand can be determined in several contact types: Pi-orbital (PO), alkyl-pi (Ak), H-donor (HD), H-acceptor (HA), and sulfur bond (SF). Figure 4 revealed that L74, V101, V486, C489 and F488 interacted with most of the molecules in this group and hence may be important residues for ligand-binding to ES.

#### 2.2.2. α-Pinene Group

There were 10 compounds in the α-pinene group. α-Pinene, the representative compound of the α-pinene group (MWGS value 0.47) was used for the predicted target protein in PharmMapper (Appendix A). The most popular target protein of α-pinene group was vitamin D binding protein (DBP, PDB ID is 1J78) [27], which has many important functions, containing and transporting vitamin D3 metabolites, binding the globular actin, and transferring fatty acids to functions in the immune system. 

Figure 5A shows the LUMO orbits of α-pinene. It can be seen that the LUMO was expressed at the C=C group, which can gain electrons more easily and become more chemically active. Figure 5B,C shows the binding pose of ten compounds to DBP, and the representative compound of α-pinene group binding to the target protein. From Table 1, it can be seen that 10 compounds had interaction with the pi-orbital of F36 and alkyl-pi interaction with V88 and M107, respectively. Hence, F36, V88, and M107 played important roles in α-pinene group compounds’ binding to DBP.

#### 2.2.3. γ-Maaliene Group

There were nine compounds in the γ-maaliene group. γ-Maaliene, the representative compound of the γ-maaliene group (MWGS value is 0.42, see Appendix A), was used as the ligand to predict target protein with PharmMapper. The most popular target protein of the γ-maaliene group was kinesin-like protein KIF11 (KSP, PDB ID is 2FKY), a motor protein required for establishing a bipolar spindle during mitosis [28]. KSP inhibitors have potential as general antiproliferative agents useful for the treatment of cancer. 

Figure 6A shows LUMO orbits of the γ-maaliene. It can be seen that LUMO was expressed at the C=C group between C6 and C7, which can gain electrons more easily and become more chemically active. Figure 6B,C show the binding pose of nine compounds to KSP, and the γ-maaliene, the representative compound of the γ-maaliene group, binding to KIF11. Table 2 shows I36, P137, L214, and R218, which play important hydrophobic roles for compounds of γ-maaliene group.

## 3. Discussions

We tested the prepared bitters dataset (Appendix A) from BitterDB (https://bitterdb.agri.huji.ac.il/dbbitter.php, accessed on 29 August 2022) using threshold CSN (THR CSN) and t-SNE dimensionality reduction CSN (t-SNE CSN), and results are shown in Figure 7. The Python script is available in the supplementary data. We observed that some molecules in the traditional THR CSNs were highly aggregated, but molecules with a generally high similarity in a large central area of the CSN were not subdivided. Molecules were well allocated to suitable locations in space according to their structural characteristics in t-SNE CSN. More importantly, the molecules in the center of the space were well clustered according to whether they were connected and how far apart they were positioned. We were then able to set the edge length threshold, to separate the group spatially.

t-SNE CSN is essentially a combination of two different representations (t-SNE and DiceSimilarity) of the 1024-bit Morgan fingerprint. Morgan fingerprinting is very suitable for terpenoids with complex topology in their essential oils, and our method can fully mine the information of the Morgan fingerprint. However, since t-SNE cannot deal with spatial discontinuity in two-dimensional space, wit is possible to obtain many molecules with higher similarity assigned to the opposite side of the CSN. This requires the design of algorithms, such as those incorporating machine learning.

Therefore, compared with THR CSN, t-SNE CSN has the advantage that a large number of aggregated molecules can be further subdivided after the introduction of coordinates, which helps to unearth more compound groups.

## 4. Materials and Methods

### 4.1. Cluster Analysis

Firstly, we determined the CSN edges. The SMILES structural formulas of 104 compounds (three isomers added) were queried on PubChem, based on the compound names inferred from the peak time of GC-MS [15,16]. The Morgan fingerprints of the molecules [29] were extracted using Rdkit [30,31,32], where the fingerprint radius was set to two. Fingerprint similarity uses Dice coefficient, and the calculation formula is as follows:(1)$$\mathrm{DiceSimilarity}\left(a,b\right)=\frac{2\times \left(a\cap b\right)\;}{a\;+\;b}$$
where a and b are the substructure features of the two molecules, respectively.

We calculated fingerprint similarity among 104 compounds and generated hierarchical clustering and correlation heatmaps with scikit-learn [33,34]. According to the similarity data, the similarity threshold was set to 0.45. The similarity of two molecules greater than the threshold forms an edge in the chemical space network, but the relative position of the compounds cannot be determined at this stage.

Next, we determined the node coordinates of CSN. Therefore, t-SNE [35,36] was used to reduce the dimensionality of the Morgan fingerprints of 104 compounds in order to obtain their relative spatial positions. To unify the number of bits of the fingerprint vector, the ECFP fingerprint [37] was generated using the explicit bitvectors method during dimension reduction; the number of bits was set to 1024, and the radius was two.

Then, we built a chemical space network, constructed according to the node coordinates and edge connections generated in the previous two steps.

Finally, we clustered and identified the representative compounds for each group. According to whether there were edge connections between molecules and whether the positions were close, nine groups of compounds were determined manually. The mean within-group similarity (MWGS) was calculated for the molecules of each group:(2)$$\mathrm{MWGS}\left(S_{i},n\right)=\frac{\left(\sum_{i=1}^{n}S_{i}\right)-1}{n}$$
where *n* is the number of molecules of the group, and is the similarity between the molecule and the *i*-th molecule in the group.

The compound with the largest MWGS in the group was considered the representative compound of the group.

In short, 91 of the 104 compounds participated in the generation of the network, and finally 73 of them were formed into nine groups, according to edge information and coordinate information (see Figure 8).

### 4.2. Group Docking and Structure Fingerprint Analysis

Using OpenBabel [38,39] to convert the SMILES formula of each group of representative compounds into mol2 format, we returned to PharmMapper [23,24] to predict the target. Then, we applied Biovia Discovery Studio to dock all molecules of each group with the corresponding target proteins. Molecular fingerprints were extracted using a Python script based on the docking results from Discovery Studio.

### 4.3. Quantum Chemical Calculations

The quantum chemical calculations were carried out using the B3LYP function [40,41,42,43] implemented in the Gaussian 09 program at the 6–31 G* set [44,45]. Frequency calculations were performed to obtain free energy corrections at 298.15 K and 1atm pressure. Multiwfn [46,47], a multifunctional program for wave function analysis of quantum chemical calculation results, was used to analyze the weak interaction of the ligands. The number of grids was set to 200 × 200 × 200 in three-dimensional space.

The 5000 frames of trajectories were extracted to average the density. To analyze traditional H-bond occupancy, the angle and distance between the donor and acceptor were set to 35°and 3.5 Å, respectively.

## 5. Conclusions

In this study, 104 compounds from essential oil in sunflower receptacles were mapped and grouped in our designed chemical space network (t-SNE CSN). The results indicated that these compounds can be divided into nine groups according to their MWGS value. PharmMapper was utilized to identify the target protein of the three families with the most members, i.e., the d-limonene, α-pinene, and γ-maaliene groups. The binding modes of the ligands of the three families to the target protein were indicated using structure fingerprint analysis. The active center of the same type of compounds was determined by quantum chemical calculation. 

## Figures and Tables

**Figure 1 ijms-23-10169-f001:**
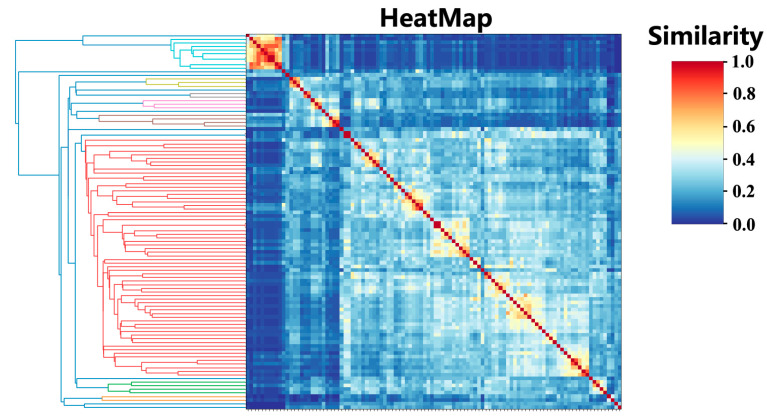
Similarity heatmap for 104 compounds. Where the similarity between molecules is greater, the color is red, and the lower the similarity, the bluer the color. The left part of the figure represents hierarchical clustering calculated according to the similarity values.

**Figure 2 ijms-23-10169-f002:**
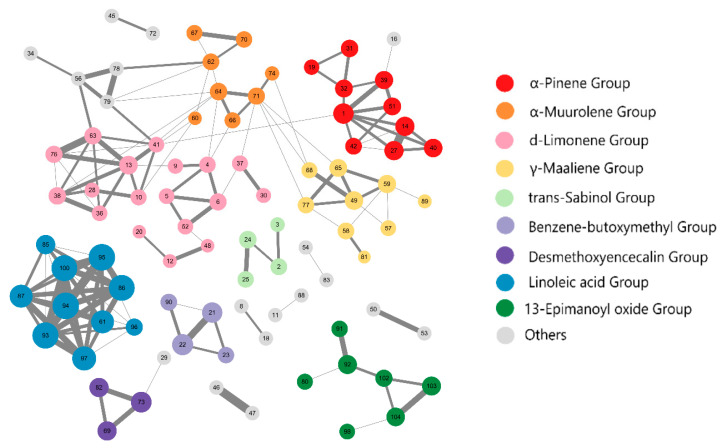
Chemical space network of 91 compounds. Nine groups with more than two molecules are colored, other groups with greater distance or less than three molecules are shown in gray.

**Figure 3 ijms-23-10169-f003:**
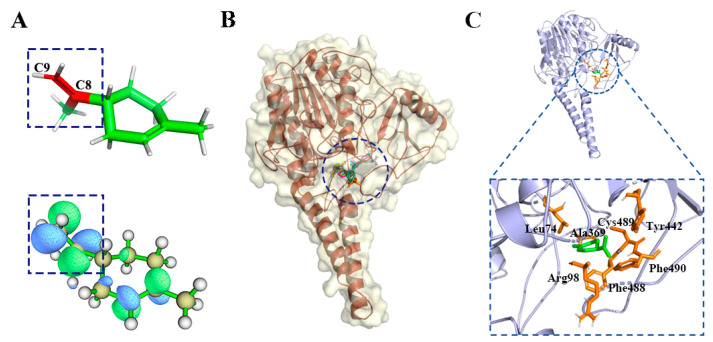
(**A**) LUMO orbit of d-limonene. (**B**) d-limonene group compounds docking with ES. (**C**) Active residues around d-limonene docking with ES.

**Figure 4 ijms-23-10169-f004:**
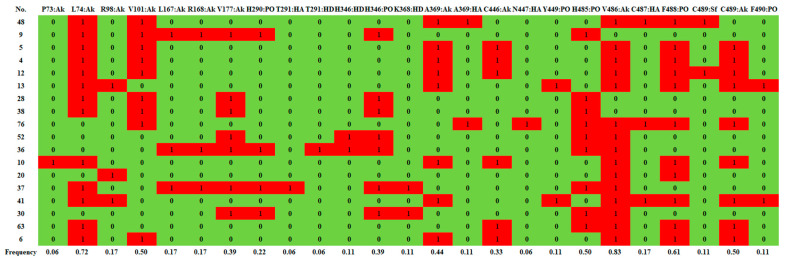
Structure fingerprint of d-Limonene group. PO, pi-orbitals; Ak, alkyl; HD, H-donor; HA, H-acceptor; Sf, sulfur.

**Figure 5 ijms-23-10169-f005:**
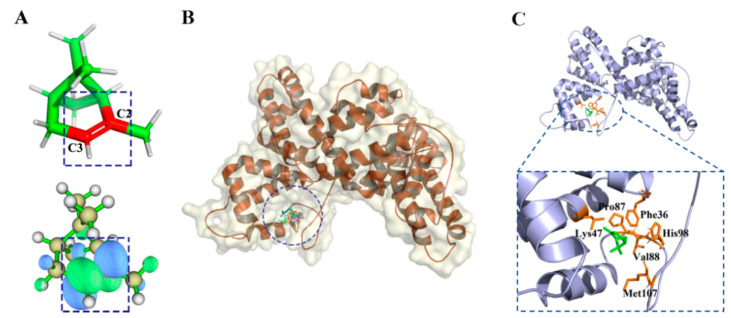
(**A**) LUMO orbit of α-Pinene. (**B**) α-Pinene group compounds docked to DBP. (**C**) Active residues around α-Pinene binding to DBP.

**Figure 6 ijms-23-10169-f006:**
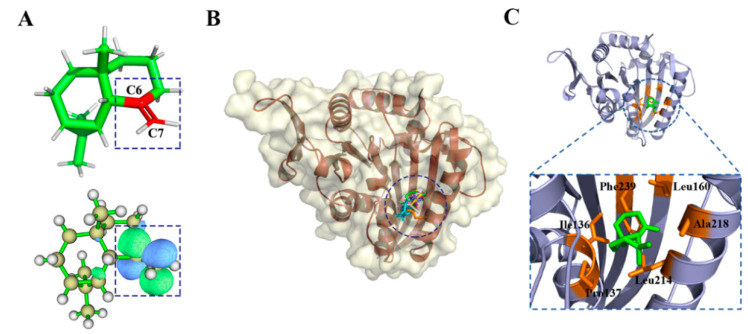
(**A**) LUMO orbit of γ-Maaliene; (**B**) γ-Maaliene group compounds docking with KIF11; (**C**) Active residues around γ-Maaliene binding to KIF11.

**Figure 7 ijms-23-10169-f007:**
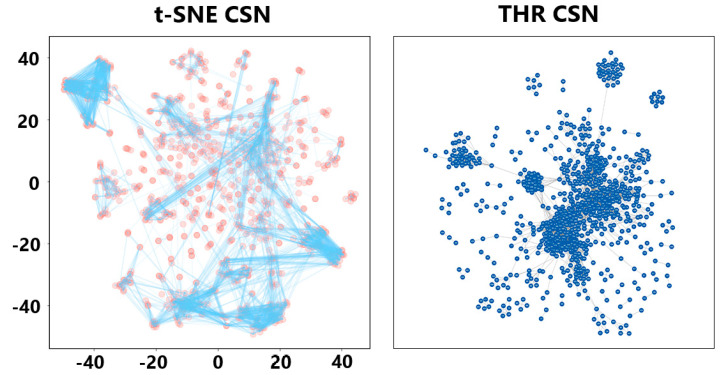
t-SNE CSN and THR CSN for the bitters dataset. Molecules in the t-SNE CSN are represented by red circles, and edges with a similarity greater than 0.45 are indicated in blue. In the THR CSN, molecules and edges are represented as blue circles and black lines, respectively.

**Figure 8 ijms-23-10169-f008:**
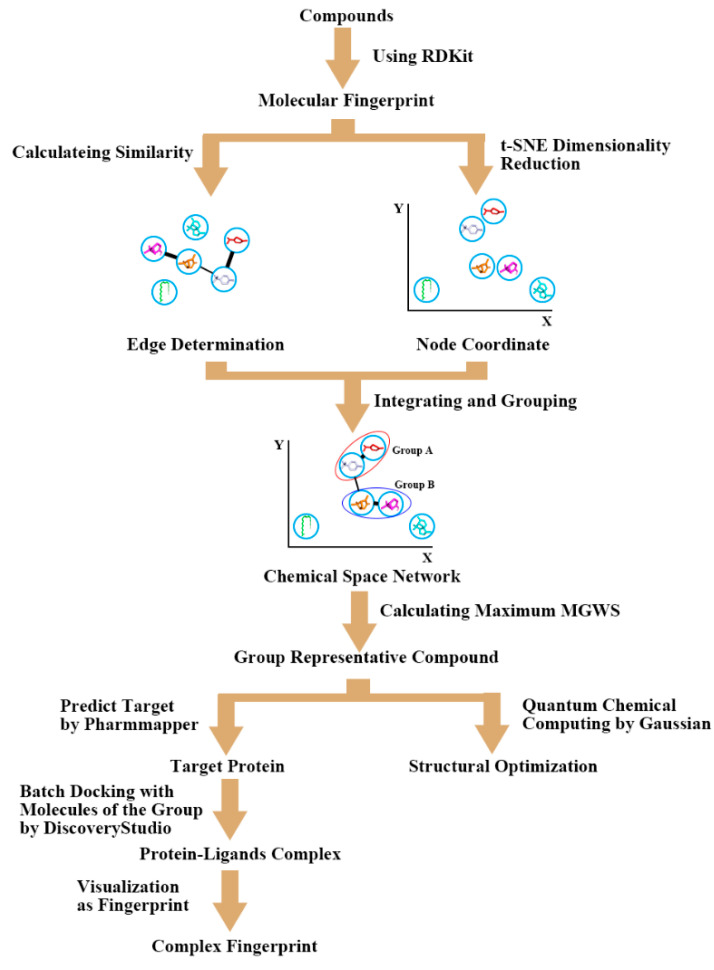
The workflow of the study methods.

**Table 1 ijms-23-10169-t001:** Fingerprint of the α-pinene group.

No.	F36:PO	L47:Ak	S76:HD	S79:HD	P87:HA	P87:Ak	V88:Ak	H89:HD	H89:PO	M107:Ak	L110:Ak
51	1	0	0	1	1	0	1	0	1	1	0
19	1	1	0	0	0	0	1	1	0	1	0
31	1	0	0	0	0	0	1	0	1	1	1
14	1	1	0	0	0	1	1	0	1	1	0
39	1	0	0	1	0	0	1	0	1	1	0
40	1	0	0	1	0	0	1	0	1	1	0
42	1	1	1	0	0	1	1	0	1	1	0
27	1	0	0	1	0	0	1	0	1	1	1
32	1	0	0	0	0	0	1	0	1	1	1
1	1	1	0	0	0	1	1	0	1	1	0

PO, pi-orbitals; Ak, alkyl; HD, H-donor; HA, H-acceptor.

**Table 2 ijms-23-10169-t002:** Fingerprint of the γ-maaliene group.

No.	E116:HA	R119:Ak	I136:Ak	P137:Ak	L160:Ak	L172:Ak	Y211:PO	L214:HA	L214:Ak	E215:HD	A218:Ak
59	0	0	1	1	1	0	0	0	1	0	1
68	0	0	1	1	1	0	0	0	1	0	1
81	0	0	1	1	1	0	0	0	1	1	1
58	0	0	1	1	1	0	0	0	1	0	1
65	0	0	1	1	0	0	0	0	1	0	1
89	0	0	1	1	1	1	0	1	1	0	1
49	0	0	1	1	1	0	0	0	1	0	1
57	0	1	1	1	0	0	1	0	1	0	0
77	1	0	1	1	0	0	1	0	1	0	0

PO, pi-orbitals; Ak, alkyl; HD, H-donor; HA, H-acceptor.

## Data Availability

MDPI Research Data Policies.

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
