# Peer review of "Clustering Analysis, Structure Fingerprint Analysis, and Quantum Chemical Calculations of Compounds from Essential Oils of Sunflower (Helianthus annuus L.) Receptacles"

_ijms, 2022, doi:10.3390/ijms231710169_

Round 1

Reviewer 1 Report

Sunflower, Heliantus annuus L., belongs to the Compositae family. Worldwide, it holds the second place, after soy, among oil plants. However, while the seeds are used to make the oil, the receptacles represent waste material. On the contrary, the receptacles can represent a source of bioactive compounds, in accordance with the current principles of green agriculture and waste recovery. New experimental approaches were used in this study (cluster analysis, reverse docking, structure fingerprint and quantum chemical calculation analysis) for the characterization of compounds present in sunflower receptacles, in order to obtain more scientific information for their use.

The topic dealt with based on the enhancement of waste material from the agro-food sector, such as sunflower receptacles, is current and falls within the aims and scope of the Journal.

The data obtained through new experimental approaches allowed to obtain different information on the bioactive compounds contained in the receptacles.

However, the text requires some changes as follows:

Figures 1-7: The figures should be understandable without the help of the text. Authors should modify and/or integrate the caption of all figures to make them easier to interpret.

Tables 1 and 2: Same comment also for the 2 tables. Furthermore, it is necessary to add some explanatory notes.

Line 128: Authors should check the text and make acronyms explicit the first time they are cited. For example: LUMO (lowest unoccupied molecular orbital) and where necessary, provide a brief explanation of the meaning. For example: LUMO orbitals are important both for the establishment of the chemical bond and in the sphere of spectroscopy.

Author Response

Comment 1 Sunflower, Heliantus annuus L., belongs to the Compositae family. Worldwide, it holds the second place, after soy, among oil plants. However, while the seeds are used to make the oil, the receptacles represent waste material. On the contrary, the receptacles can represent a source of bioactive compounds, in accordance with the current principles of green agriculture and waste recovery. New experimental approaches were used in this study (cluster analysis, reverse docking, structure fingerprint and quantum chemical calculation analysis) for the characterization of compounds present in sunflower receptacles, in order to obtain more scientific information for their us. The topic dealt with based on the enhancement of waste material from the agro-food sector, such as sunflower receptacles, is current and falls within the aims and scope of the Journal.

The data obtained through new experimental approaches allowed to obtain different information on the bioactive compounds contained in the receptacles.

However, the text requires some changes as follows:

Figures 1-7: The figures should be understandable without the help of the text. Authors should modify and/or integrate the caption of all figures to make them easier to interpret.

Answer: Thank you for your kind comment. We have revised the figure caption. Please see the revised version.

Comment 2 Tables 1 and 2: Same comment also for the 2 tables. Furthermore, it is necessary to add some explanatory notes.

Answer: Thank you for your kind comment. We have revised the Table notes. Please see the revised version.

Comment 3 Line 128: Authors should check the text and make acronyms explicit the first time they are cited. For example: LUMO (lowest unoccupied molecular orbital) and where necessary, provide a brief explanation of the meaning. For example: LUMO orbitals are important both for the establishment of the chemical bond and in the sphere of spectroscopy.

Answer: Thank you for your kind comment. We have rewritten this part. Please see the revised version.

     In page 4, line 147-151 listed as fellow:

LUMO (lowest unoccupied molecular orbital) is important both for the establishment of the chemical bond and in the sphere of spectroscopy.It depends on all of the coordinates of the system, which provides a more efficient sampling method than a geometrical reaction coordinate to better reflect the activities of the compounds [41].

Reviewer 2 Report

The manuscript "The Clustering Analysis, Structure Fingerprint Analysis and Quantum Chemical Calculations of Compounds from the Essential Oils of Sunflower (Helianthus annuus L.) Receptacles" is an interesting study of sunflower. 

I suggest a major review including a separate discussion section or merged title "3 Result and Discussion".

Some paragraphs are too short and must be merged with the main text to improve the writing quality; others must be rewritten to avoid numbers such as the "first word" in the phrase:

1) Line 61, 2.1. Cluster Analysis: Maybe change to "The molecular similarity with each other  are shown for 104 compounds (Figure 1)" instead of "104 molecular similarities with each other are shown in Figure 1."

2) LIne 77-78 must be merged and rephrasing is needed to improve it. Punctuation is missing after "Linoleic acid group is mainly long chain compounds" (line 78).  My suggestion:

"Next, we analyzed the structural characteristics of each group classification. The linoleic acid group is mainly long chain compounds. The α-pinene group compounds are mainly bicyclic monoterpenes..."

Line 91 The γ-maalineen... intead of "γ-maalineen"

Line 106: Please correct "oxygen" instead of ox-ygen

Line 126: maintaining instead of "rmaintaining"

Line 166: correct 2.2.3.γ-. maaliene group to "2.2.3. The γ-maaliene group"

In section 3.2. Cluster Analysis the author claimed that "names inferred from the peak time of GC-MS" Are these compounds described previously for the sunflower? Please indicate the references in the main text.

Line 198. avoid paragraph or initiate the phrase with Where instead of where. The same in line 217: "n is the..."

Conclusion

The authors claimed that "104 compounds were identified by cluster analysis", but the identifications are inferred from CG-MS as indicated Material and methods (3.2. Cluster Analysis). The authors must be more clear about that.

Author Response

Comment 1 The manuscript "The Clustering Analysis, Structure Fingerprint Analysis and Quantum Chemical Calculations of Compounds from the Essential Oils of Sunflower (Helianthus annuus L.) Receptacles" is an interesting study of sunflower. 

I suggest a major review including a separate discussion section or merged title "3 Result and Discussion".

Answer: Thank you for your kind comment. We have rewritten this part. A separate discussion was shown in revised version. Please see the revised version.

In page 7-8, the revised version is present as fellow.

  1. Discussions

We tested the prepared bitters dataset (Table S13) from BitterDB (https://bitterdb.agri.huji.ac.il/dbbitter.php) using threshold CSN (THR CSN) and t-SNE dimensionality reduction CSN (t-SNE CSN), and result is shown in Figure 7. The Python script is available in Supplementary Data. We can see that some molecules in the traditional THR CSNs are highly aggregated, but the molecules with a generally high similarity in a large area in the middle of the CSN are not subdivided. Molecules are well allocated to suitable locations in space according to its own structural characteristics in t-SNE CSN, more importantly, the molecules in the center of the space are well clustered according to whether they are connected and how far apart. Then we can set the edge length threshold to separate the group in space.

t-SNE CSN is essentially a combination of two different representations (t-SNE and DiceSimilarity) of the 1024-bits Morgan fingerprint. Morgan fingerprint is very suitable for terpenoids with complex topology in essential oils, and our method can fully mine the information of Morgan fingerprint. However, since t-SNE cannot deal with spatial discontinuity in two-dimensional space, we can easily get many molecules with higher similarity to be assigned to the opposite side of CSN. This requires the design of algorithms, such as incorporating machine learning. Therefore, compared with THR CSN, t-SNE CSN has the advantage that a large number of aggregated molecules can be further subdivided after the introduction of coordinates, which will help us unearth more compound groups.

Comment 2 Some paragraphs are too short and must be merged with the main text to improve the writing quality; others must be rewritten to avoid numbers such as the "first word" in the phrase:

1) Line 61, 2.1. Cluster Analysis: Maybe change to "The molecular similarity with each other  are shown for 104 compounds (Figure 1)" instead of "104 molecular similarities with each other are shown in Figure 1."

2) Line 77-78 must be merged and rephrasing is needed to improve it. Punctuation is missing after "Linoleic acid group is mainly long chain compounds" (line 78).  My suggestion:

"Next, we analyzed the structural characteristics of each group classification. The linoleic acid group is mainly long chain compounds. The α-pinene group compounds are mainly bicyclic monoterpenes..."

Line 91 The γ-maalineen... intead of "γ-maalineen"

Line 106: Please correct "oxygen" instead of ox-ygen

Line 126: maintaining instead of "rmaintaining"

Line 166: correct 2.2.3.γ-. maaliene group to "2.2.3. The γ-maaliene

group"

Answer: Thank you for your kind comment. We have revised these parts that the reviewer’ refer to. Please see the revised version.

In page 2 line 52-68, the revised version is present as fellow.

Chemical Space Network (CSN) maps chemical molecules into a visual space according to some of their characteristics, such as structure. CSN was initially designed as a coordinate-free threshold networks using the Tanimoto coefficient as a continuous similarity measure[17]. However, it can only reflect the connection between molecules and cannot reflect the relative distance of molecules in space. Several studies propose more novel CSNs like TV-CSN[18] and Kamada-Kawai network[19], the latter tried to introduce coordinates into the CSN. Compared to the traditional threshold CSN, here we added t-SNE dimensionality reduction to determine the spatial coordinates to adapt to the components clustering of mixtures with large differences in composition, such as essential oils.

Protein-ligand interaction fingerprints (IFPs) are binary 1D representations of the 3D structure of protein-ligand complexes encoding the presence or absence of specific interactions between the binding pocket amino acids and the ligand. IFPs have successfully applied for post-processing molecular docking results for G Protein-Coupled Receptor (GPCR) ligand binding mode prediction and virtual ligand screening[20].

In page 2 line 69-72, the revised version is present as fellow.

Over 100 compounds were clustered by mapping into a CSN and selected representative compound within each group. The target of moelcules of each group was identified using reverse docking to the group representative compound.

In page 2 line 79-85, the revised version is present as fellow.

  The molecular similarities with each other are shown for 104 compounds (Figure 1). Due to the complex composition of essential oils, the similarity between molecules is generally low. But it is still obvious that some molecules are similar, such as long-chain compounds (the upper left corner of the heatmap). There is a whitish area in the heatmap (hierarchical clustering is shown in red), which is a large cluster of endocyclic compounds.

In page 2 line 94-104, the revised version is present as fellow.

According to the edges data in Figure 1 and the node coordinate data in Table S2, We group these molecules based on the similarity greater than 0.45 and node location closer. After filtering out groups whose total number of molecules is less than 3, 73 of 91 compounds clustered into nine classes, see Figure 2 and Table S3-S11. We can see that many related molecules are not grouped into one group because they are too far apart, which is exactly what we expected to refine the groups. α-Pinene group, d-limonene group, α-muurolene group and γ-maalineen group are more or less connected, traditional methods have difficulty distinguishing these molecules that link multiple groups. But after introducing the coordinates obtained by dimensionality reduction, the four groups are visually separated that preliminarily proves that our method is feasible in the essential oils with large differences in components.

Comment 3 In section 3.2. Cluster Analysis the author claimed that "names inferred from the peak time of GC-MS" Are these compounds described previously for the sunflower? Please indicate the references in the main text.

Answer: Thank you for your kind comment. We have cited Ref. 18 and Ref. 19 there. Please see the revised version.

Comment 4 Line 198. avoid paragraph or initiate the phrase with Where instead of where. The same in line 217: "n is the..."

Answer: Thank you for your kind comment. We have rewritten this sentence. Please see the revised version.

Conclusion

Comment 5The authors claimed that "104 compounds were identified by cluster analysis", but the identifications are inferred from CG-MS as indicated Material and methods (3.2. Cluster Analysis). The authors must be more clear about that.

Answer: Thank you for your kind comment. We are so sorry that we made a mistake. In fact, 104 compounds were identified from CG-MS, please see the revised version.

Reviewer 3 Report

The problems studied in the work are interesting, but somehow the dominant mathematical data processing, calculations, application of statistical techniques and tools..., in relation to concrete experimental work and research.

Author Response

Reviewer 3

Comment 1The problems studied in the work are interesting, but somehow the dominant mathematical data processing, calculations, application of statistical techniques and tools..., in relation to concrete experimental work and research.

Answer: Thank you for your kind comment. In this study, we are doing theoretical calculations. In the next work, the experimental from these calculations can be done.

Round 2

Reviewer 1 Report

The authors responded satisfactorily to the requests made, on the basis of which they modified and integrated their work.

Reviewer 2 Report

The revised version of the manuscript addressed all issues and will be published in its present form.

Reviewer 3 Report

Thank you for the explanation provided.
I wish you much success and I offer you full support in realizing your future research plans.